# Penile Cancer Profile in a Central European Context: Clinical Characteristics, Prognosis, and Outcomes—Insights from a Polish Tertiary Medical Center

**DOI:** 10.3390/cancers17132140

**Published:** 2025-06-25

**Authors:** Mateusz Czajkowski, Michał Falis, Agata Błaczkowska, Agnieszka Rybarczyk, Piotr M. Wierzbicki, Jakub Gondek, Marcin Matuszewski, Oliver W. Hakenberg

**Affiliations:** 1Department of Urology, Medical University of Gdańsk, Mariana Smoluchowskiego 17 Street, 80-214 Gdańsk, Poland; m.falis@gumed.edu.pl (M.F.); kba@gumed.edu.pl (J.G.); matmar@gumed.edu.pl (M.M.); 2Division of Radiology Informatics and Statistics, Medical University of Gdańsk, Tuwima 15 Street, 80-210 Gdańsk, Poland; agata.blaczkowska@gumed.edu.pl; 3Department of Histology, Medical University of Gdańsk, Dębinki, 80-211 Gdańsk, Poland; agnieszka.rybarczyk@gumed.edu.pl (A.R.); pwierzb@gumed.edu.pl (P.M.W.); 4Department of Urology, Jena University Hospital, 07747 Jena, Germany; oliver.hakenberg@med.uni-jena.de

**Keywords:** penile cancer, phimosis, smoking, overweight, obesity, HPV, prognosis

## Abstract

This study examined the prevalence of well-established clinical and lifestyle characteristics associated with penile cancer and clinical outcomes for penile cancer in 153 patients treated at a Polish medical center. Phimosis, smoking, obesity, and rural residence were frequently observed, whereas HPV infection was less prevalent. Smoking significantly increased the risk of mortality and decreased overall survival. Conversely, HPV infection demonstrated a trend toward improved survival. These findings underscore the importance of early detection and addressing modifiable factors to enhance outcomes, particularly in Central Europe, where data on penile cancer remain limited.

## 1. Introduction

Penile cancer is a rare malignancy in European and North American countries, accounting for less than 1% of malignant neoplasms [1]. In contrast, some countries in Asia, Africa, and South America have the highest incidences of penile cancer globally [2,3]. According to the National Cancer Registry Report for 2021, the incidence and mortality rates of penile cancer in Poland were low, at 259 and 138, respectively [4]. Nevertheless, due to the under-registration of penile cancer cases, its true incidence rate is likely higher. Reasons for this under-registration are possibly the lack of histopathological examination for every foreskin specimen following circumcision and the failure to obtain representative specimens prior to ablative treatment modalities, e.g., laser therapy [5].

Notably, Poland is one of the few European countries where an increase in penile cancer mortality has been observed, possibly associated with a lack of early symptom identification [5]. This phenomenon may be attributed to the very low rate of circumcision in Poland (0.11%), which is associated with a higher incidence of phimosis [6]. The necessity for the earliest possible diagnosis of penile cancer remains uncontested, since the detection of penile cancer at stages I and II results in five-year survival rates, after surgical treatment, of up to 85%. In contrast, when penile cancer is identified at stages III and IV, the five-year survival rate drops significantly, to 59%, and further declines to 11% [7]. Early detection permits organ-sparing surgery, which, despite more frequent local recurrences, provides superior aesthetic outcomes and the preservation of penile function, while maintaining long-term overall and metastasis-free survival at the same level as classic penectomy [8].

The etiology of penile cancer is characterized by a complex interplay among various risk factors such as phimosis, tobacco use, excess body weight, and human papillomavirus (HPV) infection [9]. Phimosis creates an environment that leads to chronic inflammation, potentially leading to cellular alterations [10,11]. Tobacco use increases the risk of penile cancer through systemic effects and local tissue changes [12]. Human papillomavirus (HPV) infection plays a crucial role in the development of penile cancer by interfering with normal cell cycle regulation [2].

However, the relative importance of these risk factors can vary significantly across diverse geographical and socioeconomic contexts. In regions with higher rates of circumcision, phimosis-related risks are less relevant [10]. Conversely, in regions with inadequate sanitation, hygiene-related factors may have greater significance. Socioeconomic status can influence the exposure to penile cancer risk factors, with lower-income populations potentially facing higher risks owing to their limited access to healthcare, education, and preventive measures [3]. Furthermore, cultural practices and beliefs can impact the prevalence of risk factors and healthcare-seeking behaviors [13]. Understanding these nuanced interactions between risk factors and their contextual variations is crucial for developing targeted prevention strategies and improving outcomes in penile cancer management across diverse populations.

Owing to the rarity of penile cancer, there is insufficient information regarding the prevalence of risk factors and their association with prognosis in different countries. This deficiency is particularly evident in Central Europe, specifically Poland, as most studies originate from South America or Africa. A comprehensive analysis of well-established clinical and lifestyle characteristics associated with penile cancer and their impact on prognosis in Poland is important because of the elevated mortality rates associated with penile cancer and for the implementation of adequate interventions [5].

This study aimed to address this gap by characterizing the clinical presentation of penile cancer in a Polish patient cohort, determining the prevalence of selected clinical and behavioral characteristics, evaluating long-term surgical outcomes, and examining their associations with survival.

## 2. Materials and Methods

This retrospective study was conducted from October 2011 to October 2024 at a single tertiary medical center (Department of Urology, Medical University of Gdansk) and included 153 patients who underwent surgical treatment for penile cancer during this period. This study was approved by an independent ethics committee (Bioethics Committee for Scientific Research at the Medical University of Gdańsk; Decision No. NKBBN/168/2023). The inclusion criteria were as follows: individuals ≥18 who had undergone surgery for penile cancer and consented to the proposed treatment method, specifically those with squamous cell carcinoma (SCC) of the usual type or verrucous squamous cell carcinoma. The exclusion criteria encompassed the refusal of the proposed treatment method and the presence of histopathological types of penile cancer not specified in the inclusion criteria. Demographic and histopathological data, including the types of penile cancer classified according to the 2020 WHO classification and pathological staging using the *TNM Classification of Malignant Tumors*, 8th edition, were collected [14,15]. Given the extended duration of the study and potential inconsistencies in pathological classification, all histopathological specimens were re-evaluated by an experienced uropathologist from December 2024 to January 2025.The prevalence of human papillomavirus (HPV) in penile cancer specimens was assessed using p16 immunohistochemistry, a useful surrogate marker for transcriptionally active high-risk HPV, though it is neither fully specific nor fully sensitive. According to a recent systematic review conducted by Olesen et al., among HPV DNA-positive penile cancer tumors, 79.6% were also p16 positive. However, among HPV DNA-negative penile cancer tumors, 18.5% still exhibited p16 positivity [16].

A comprehensive, self-administered patient survey (Appendix A) was conducted to identify the presence of selected clinical and behavioral characteristics. An age of >71 years (4th quartile) was considered as advanced age. A body mass index (BMI) ≥ 25–29.9 kg/m^2^ was categorized as overweight, while a BMI ≥ 30 kg/m^2^ was classified as obesity. The size of the residential area, the number of male inhabitants, and the distance from the referral center were obtained. The place of residence was categorized according to its size: rural areas, small urban centers (less than 20,000 inhabitants), medium-sized city (20,000–100,000 inhabitants), and large metropolitan areas (more than 100,000 inhabitants). A distance from the referral center of >147 km (4th quartile) was considered as “far”.

## 3. Statistical Methods

Correlations between clinical and behavioral characteristics were evaluated using Pearson’s correlation coefficients and corresponding *p*-values to determine statistical significance. A heatmap of the correlation matrix was generated to visualize the relationships, with coefficients ranging from −1 to +1. A significance level of α = 0.05 was adopted for all statistical tests.

A multivariate Cox proportional hazards regression model was employed to assess the impact of clinical factors, including age (median-centered), smoking, HPV, lymph node stage, and tumor grade on survival outcomes. For each covariate, statistical power was estimated based on the Wald z-statistic (two-sided α = 0.05), following standard methods for regression models. The number of included covariates was limited according to the standard events-per-variable (EPV) rule (>10), considering 48 deaths in the study cohort. Variables with low statistical power and non-significant univariable associations were excluded to reduce the risk of overfitting. As such, the pT stage was not included in the final model. Proportional hazards assumptions were tested using Schoenfeld residuals and were not violated for any covariate (all *p* > 0.05). Additionally, the global test for the proportional hazards assumption was not violated (*p* = 0.1692). Adjusted survival curves for the smoking and non-smoking groups were generated using the Cox proportional hazards model, with all other covariates held at their median values. The curves illustrate differences in survival probabilities over time between the two groups.

A Kaplan–Meier analysis was performed to assess overall survival (OS) and cancer-specific survival (CSS. The Tarone–Ware test was applied for additional survival curves to account for intersecting survival curves and to better capture differences between groups.

Across all methods, Python libraries, such as matplotlib, numpy, pandas, scipy, seaborn, lifelines, and statsmodels, were extensively utilized for data processing, statistical modeling, and visualization. Missing values were excluded from the dataset. All statistical analyses were conducted using Python, version 3.11 (Python Software Foundation, Wilmington, DE USA). Data preprocessing and management were performed using pandas (version 2.2.3) and numpy (version 2.1.3). Statistical tests, including the Cox proportional hazards regression and Kaplan–Meier survival analysis, were implemented using lifelines (version 0.30.0) and statsmodels (version 0.14.4). The visualization of data and results was performed with matplotlib (version 3.9.2) and seaborn (version 0.13.2). Missing values were excluded from the dataset.

## 4. Results

### 4.1. Patient Characteristics

The analysis presented below was based on data from 153 patients who underwent surgery for penile cancer at a single tertiary medical center over a period of 13 years (October 2011 to October 2024). Their median age was 64 years (range: 30–87); the median BMI was 29 kg/m^2^ (range: 16–38). The residential distribution of the 153 patients was as follows: 42 (27.45%) lived in rural areas; 24 (15.95%) in small urban centers; 41 (26.80%) in medium-sized cities; and 46 (30.07%) in metropolitan areas. The median distance from the referral center was 62 km (mean: 104.54; SD ± 120). The median follow-up duration was 27 months (mean: 37.61; range: 4–159), during which 48 patients died and 105 remained alive (Table 1).

### 4.2. Surgical Treatments

For surgical treatment, among the 153 patients, 20 (13.07%) underwent circumcision; 12 (7.84%) underwent wide local excision; 55 (35.95%) underwent glansectomy with reconstruction using a split-thickness skin graft; 49 (32.03%) underwent partial penectomy with reconstruction using a split-thickness skin graft; 10 (6.54%) underwent partial penectomy without reconstruction; and 7 (4.58%) underwent total penectomy. Additionally, the invasive surgical staging of inguinal lymph nodes was performed by dynamic sentinel lymph node biopsy (DSNB) in 27 (17.65%) patients; bilateral inguinal modified lymphadenectomy was performed on 82 (53.59%) patients. Forty-four (28.76%) patients underwent radical inguinal lymph node dissection, and 21 (13.73%) underwent pelvic lymphadenectomy.

### 4.3. Radiotherapy and Chemotherapy

Radiotherapy was administered to the primary tumor for only 2 (1.3%) patients; however, these patients underwent subsequent partial penectomy with reconstruction using a split-thickness skin graft due to necrosis. Additionally, adjuvant radiotherapy was performed on the inguinal region for 2 (1.3%) patients with pN2 and 4 (2.61%) patients with pN3.

Neoadjuvant chemotherapy was provided to 7 (4.6%) patients with pN3 using 4 cycles of paclitaxel, cisplatin and ifosfamide (TIP). Additionally, adjuvant chemotherapy was administrated to 15 (9.8%), 11 (7.2%), and 16 (10.46%) patients with pN1, pN2, and pN3, respectively. In terms of adjuvant therapy regimens, TIP was administered to 32 (20.92%) patients, while cisplatin and paclitaxel (PF) were given to 10 (6.54%) patients. Immunotherapy based on Pembrolizumab was employed as a second- or third-line treatment for 8 (5.23%) patients.

### 4.4. Histopathological Findings

A total of 147 (96.08%) patients were diagnosed with the usual type of squamous cell carcinoma, while 6 (3.92%) presented with verrucous squamous cell carcinoma. The results of grading were as follows: penile intraepithelial neoplasia (PeIN) III (*n* = 22; 14.38%); well-differentiated G1 (*n* = 35; 22.88%); moderately differentiated G2 (*n* = 68; 44.44%); and poorly differentiated G3 (*n* = 28; 18.30%). The pathological staging (TNM) was as follows: 22 (14.38%) Tis; 45 (29.41%) pT1a; 20 (13.07%) pT1b; 38 (24.84%) pT2; 23 (15.03%) pT3; and 5 (3.27%) pT4 cases. Lymph node involvement (pN-stage) was as follows: 99 (64.71%) pN0; 20 (13.07%) pN1; 11 (7.19%) pN2; and 23 (15.03%) pN3 cases. Distant metastases were observed in three patients.

HPV infection was identified using p16 immunohistochemistry in 22 (14.38%) patients.

### 4.5. Clinical and Behavioral Characteristics

The associations between the clinical and behavioral characteristics associated with penile cancer were investigated using the data collected through the self-administered patient survey (Appendix A).

The clinical and behavioral characteristics identified in our cohort were phimosis, residence in small agglomerations (rural areas and small urban centers), and an increased BMI (being overweight (BMI ≥ 25–29.9 kg/m^2^) as well as obese (BMI ≥ 30 kg/m^2^)); these features were observed in 72 (47.06%), 66 (43.14%), 66 (43.14%), and 62 (40.52%) patients, respectively. It is important to note that all patients diagnosed with phimosis in our study exhibited complete phimosis (grade 5) [17]. Other identified factors were smoking for 59 (38.56%) patients, residing far from the referral center for 40 (26.14%) patients, advanced age for 39 (25.49%) patients, HPV infection for 22 (14.38%) patients, agricultural occupation for 15 (9.8%) patients, beekeeping occupation for 13 (8.50%) patients, and psoriasis for 3 (1.96%) patients (Figure 1). Evidently, some patients presented several of these factors: 49 (32.07%) patients exhibited both phimosis and residence in small agglomerations. Furthermore, advanced age (>71 years) and an increased BMI were observed in 32 patients (20.92%) (Figure 1).

A correlation analysis of the individual factors showed a positive correlation between residence in small agglomerations and beekeeping (r = 0.16; *p* < 0.05). Conversely, a slight negative correlation was noted between advanced age (>71 years) and distance from the referral center (>147 km) (r = −0.27; *p* = 0.001) (Figure 2). Importantly, there was a positive correlation between phimosis and more advanced penile cancer stage (TNM) (r = 0.16; *p* = 0.045). The coefficient indicates that the presence of phimosis is likely to be associated with a slight increase in the probability of a more advanced penile cancer stage.

### 4.6. Patient Survival

In the study cohort, the stage of the primary tumor (T1–T4) had a significant impact on patient survival. The Tarone–Ware test showed no significant differences in OS between the T1 and T2 stages (*p* = 0.93); however, significant differences were observed between the T2 and T3 stages (*p* = 0.03). Moreover, patients with penile cancer at stage T4 exhibited significantly worse CSS than those with T3 stage cancer (*p* = 0.01) (Figure 3).

Similarly, no significant differences in OS were observed between the N0 and N1 stages (*p* = 0.97). However, significant differences in OS were observed between N1 and N2 stages (*p* < 0.005). Notably, there was no statistically significant difference between N2 and N3 stages (*p* = 0.54) (Figure 4).

According to tumor grade, patients with poorly differentiated (G3) features exhibited poorer survival than those with well-differentiated (G1) and moderately differentiated (G2) features (G1 vs. G2, *p* = 0.2121; G1 vs. G3, *p* = 0.0003; G2 vs. G3, *p* = 0.0038).

Penile cancer was identified as the leading cause of death, accounting for 38 (79.17%) of the total fatalities. Furthermore, other causes of mortality included heart failure (*n* = 3, 6.25%), lung cancer (*n* = 2, 4.27%), stroke (*n* = 2, 4.27%), colon cancer (*n* = 1, 2.09%), liver cirrhosis (*n* = 1, 2.09%), and COVID-19 infection (*n* = 1, 2.09%).

### 4.7. Risk of Death

To investigate the influence of clinical and lifestyle characteristics associated with penile cancer on mortality, the Cox test was employed. The model achieved a concordance index (C-index) of 0.773, indicating a moderate predictive accuracy. Among all clinical and behavioral features, only smoking was associated with a significant, two-fold higher risk of death (HR = 1.81; *p* = 0.047); the OS of smokers was significantly lower than that of non-smokers (*p* = 0.047) (Table 2). The survival curve for smokers demonstrated a more rapid decline than that of non-smokers. The median overall survival for smokers was approximately 40 months, whereas that for non-smokers was 70 months. Additionally, after 140 months, the probability of survival in the smoker group was 20% versus 40% in the non-smoker group (Figure 5). Additionally, lymph node involvement status (HR = 2.10, *p* = 0.027) and tumor grade were significantly associated with survival (HR = 1.85, *p* = 0.003).

**Variable definitions:** Age and BMI were included as continuous variables; age was centered at the median, and BMI was modeled on the natural logarithmic scale. Smoking status, HPV status, phimosis, beekeeping, psoriasis, and lymph node stage were treated as binary variables (0 = absence/no/low grade, 1 = presence/yes/high grade). Tumor grade was treated as an ordinal variable representing tumor aggressiveness, with four categories: pTis/G0 (0), G1 (1), G2 (2), and G3 (3). This variable was included to assess the association between increasing grade and survival. Residence in small agglomerations was coded as an ordinal variable: 0 = rural, 1 = small town, 2 = medium-sized town, 3 = large city.

Conversely, HPV infection reduced the risk of death by improving CSS by almost six-fold (HR = 0.15); however, this result approached the significance level, but did not reach it (*p* = 0.063) (Figure 6). Other analyzed clinical and lifestyle characteristics associated with penile cancer mentioned in Table 2 were not associated with mortality. Furthermore, although the pT stage was initially considered for inclusion in the multivariable Cox regression analysis, it was ultimately excluded due to its lack of statistical significance (HR: 1.50, 95% CI: 0.27–8.24, *p* = 0.638) and very low statistical power (0.07).

## 5. Discussion

### 5.1. Uncircumcised Males and Penile Cancer Risk: The Influence of Phimosis

Phimosis is classified as primary, mainly affecting pediatric populations, or as secondary (pathological), resulting from conditions like obesity, diabetes, and lichen sclerosus [10,18]. It affects uncircumcised males, with distribution varying across regions due to different male circumcision rates. The global prevalence of circumcised males is 37–39% [6], varying significantly across countries due to cultural and religious factors [19]. Morocco has the highest circumcision rate (99.9%), while most European countries have under 10% [6]. In Eastern and Central Europe, rates are even lower, from 0.11% in Poland to 2.3% in Ukraine [6].

Phimosis is found in 25–75% of penile cancer cases, highlighting the close association between these two pathologies [20,21]. According to Daling et al., phimosis was diagnosed in 35.2% of patients with penile cancer and only in 7.6% of a control group. Furthermore, phimosis was associated with an increased risk of invasive penile cancer (OR = 2.3), but not with an increased risk of Tis penile cancer (OR = 1.1) [22]. A similar observation was made by Tseng et al., where 35% of penile cancer cases exhibited phimosis at least two years prior to diagnosis. Furthermore, phimosis demonstrated a strong association with invasive penile cancer (OR = 16; 95% Cl = 4.5–57), while no significant correlation was observed with Tis penile cancer (OR = 0.41; 95% Cl = 0.13–1.1) [23]. According to Brinton et al., who enrolled 150 patients with penile cancer in a case–control study, 45 patients (30%) with penile cancer were not circumcised. Among the uncircumcised cases, phimosis was present in 73.3% of the penile cancer group compared with 7.2% of the controls [24]. A population-based case–control study from Denmark, where the estimated prevalence of childhood circumcision is under 2%, included 71 cases of penile cancer (53 invasive and 18 Tis), all of which were uncircumcised, compared to only 5 uncircumcised individuals among 189 controls. Phimosis was identified in 27% of the patients with penile cancer and was present in 10–11% of the patients in the control group. There was a significant correlation between phimosis and the risk of penile cancer (OR = 3.39; 95% CI = 1.62–7.11) [25]. According to Borque-Fernando et al., phimosis was present in 45.2% (*n* = 103) of penile cancer cases, whereas 30.7% (*n* = 70) were circumcised [26]. Our findings also demonstrate that phimosis is a significant clinical characteristic for penile cancer, with an incidence of 47.06% (*n* = 72) in our cohort study. Moreover, the presence of phimosis was associated with more locally advanced (T-stage) disease (r = 0.16; *p* = 0.045), which correlated with significantly poorer outcomes (Figure 3). Consistent with the circumcision rate in Poland (0.11%), all patients in our cohort were uncircumcised [6].

The hypothesis for penile cancer carcinogenesis associated with phimosis suggests that phimosis impairs hygiene, causing smegma accumulation and microbial proliferation beneath the foreskin. This chronic inflammation initiates carcinogenesis through angiogenesis and cellular proliferation [27,28,29,30]. The inflammatory cells’ mediators inhibit neoplastic cell apoptosis and accelerate tumor growth [30,31]. However, smegma itself is not considered a cause of penile carcinoma [32].

### 5.2. Elevated BMI and Penile Cancer Risk

Overweight and obesity are major healthcare challenges globally. The 2022 Obesity Report showed that 43% of adults are overweight and 16% obese, with one in eight adults having a BMI ≥ 30 kg/m^2^ [33]. Obesity accounts for 4–8% of cancer cases, with excess adipose tissue raising cancer mortality risk by 17% [34].

Excessive body weight, as assessed by elevated body mass index (BMI), is also a recognized risk factor for penile cancer. According to Aune et al., who analyzed data from the Norwegian Tuberculosis Screening Program and linked it with data from the Cancer Registry of Norway, penile cancer was diagnosed in 725 of 829,081 patients [35]; their comprehensive Norwegian population study demonstrated a significant association between excess body mass and an elevated penile cancer risk. Compared to individuals with a normal body mass index, obese subjects exhibited a 63% higher penile cancer risk. Specifically, individuals with grade 1 obesity presented a 52% greater penile cancer risk, whereas those with grade 2 obesity displayed a 228% increase in penile cancer risk [35]. Similarly, Barnes et al. identified 101 penile cancer patients with a mean BMI of 31.8 kg/m^2^. In that study, multivariate ordinal logistic regression analysis showed that each 5-unit increment in BMI was associated with a 49% increase in the probability of a higher stage at diagnosis [36]. These results were confirmed in a population-based case–control study wherein penile cancer cases exhibited a significantly higher prevalence of being overweight or obese than the controls. Furthermore, the odds ratios were 2.64 (95% CI = 1.81–386; *p* = 0.0103) and 3.24 (95% CI = 2.07–5.08; *p* = 0.0002) for penile cancer among men with overweight vs. normal weight and obesity vs. normal weight, respectively [37]. An acquired buried penis is a condition in which the accumulation of skin and/or adipose tissue in the suprapubic area envelops a penis of normal size, concealing it [38]. According to Higuchi et al., up to 87% of patients undergoing surgical treatment for a buried penis are obese [39]. Notably, a retrospective study of 150 adults with acquired buried penis revealed a 7% prevalence of penile cancer and 35% prevalence of premalignant conditions, including penile intraepithelial neoplasia (PEIN), condyloma, and lichen sclerosus [40]. This study identified overweight (*n* = 62, 40.52%) and obesity (*n* = 66, 43.14%) as predominant clinical and lifestyle characteristics for penile cancer, which is consistent with previous research. Nevertheless, our data did not indicate a correlation between an elevated BMI and advanced-stage disease or prognosis.

The mechanisms linking elevated BMI and penile cancer remain unclear. The primary hypothesis suggests that excess weight impairs penile hygiene, leading to lichen sclerosus due to occlusion and irritation from urine. This condition can cause phimosis through fibrotic ring formation [41,42]. Additionally, obesity-related buried penis prevents lesion detection [37]. The secondary hypothesis proposes that adipose tissue’s metabolic and hormonal activity [43] promotes carcinogenesis through hyperinsulinemia and insulin-like growth factor-1 (IGF-1) pathway alterations [44]. Obesity may facilitate oncogenesis through chronic low-grade inflammation and oxidative stress [44], linked to adipose tissue’s pro-inflammatory cytokines [45,46]. This inflammation contributes to cancer initiation and progression, affecting cellular transformation, survival, proliferation, invasion, and metastasis [47].

### 5.3. Rural Living and Penile Cancer Incidence

Residence in rural areas or small communities is one of the less frequently examined factors associated with penile cancer in the literature. One of the early investigations examining this association was published in 1999 by Wesseling et al. based on data from the Costa Rican Cancer Registry encompassing the years 1981–1993 [48]. A significantly higher incidence of penile cancer was seen in rural areas and Costa Rican agricultural regions, particularly coffee plantations. This was attributed to high pesticide exposure (e.g., lead arsenate and paraquat) and low socioeconomic status [48]. According to Coelho et al., among the 392 cases reported by the Maranhão referral center for penile cancer treatment between 2004 and 2014, 71% of the patients were agricultural workers and 82.1% resided in rural areas. The high incidence of penile cancer in this district was associated with a high prevalence of HPV infection, which was observed in up to 75% of patients with penile cancer [3]. This finding may be indirectly supported by the work of Pinho-França et al., who demonstrated that Maranhão has the highest cervical cancer incidence among all Brazilian regions, a malignancy for which HPV infection is also a proven risk factor [49]. The investigation conducted by Vieira et al. in Maranhão and São Luís showed that, among 116 penile cancer patients, 57% resided in rural areas and 58% worked in agriculture. In addition to the high prevalence of HPV infection, the considerable distance from major health centers and low or absent formal education likely contribute to the unique epidemiology of penile cancer in this region [50]. Similarly, in a study conducted in Nepal, Sigdel et al. enrolled 380 patients with penile cancer and found that 83.1% had an agricultural occupation and 69.6% were unable to read and write [51]. Since all these studies originated outside Europe, it is impossible to directly extrapolate these results to Central Europe. This limitation also applies to socioeconomic differences in education, sexual health awareness, rural living conditions, and agricultural practices, including variations in pesticide usage, which are influenced by the types of crops cultivated and the regulatory frameworks governing pesticide application.

Rural areas in Poland housed 15.4 million people (40.1% of total population) in 2020. Males outnumbered females in age groups up to 64 years, with the reverse trend in the 65+ group. [52]. Of the 153 patients included in this study, 42 (27.45%) resided in rural areas, and 24 (15.95%) were from small towns with populations below 20,000 inhabitants. Furthermore, only 15 participants (9.8%) were engaged in agricultural occupations. Residence in small communities may be associated with cumulative risk factors. Higher tobacco use, lower smoking cessation rates [53], lower HPV vaccination rates [54], and exposure to chemical plant protection agents [55] are potential penile cancer risk factors requiring future research in European populations. However, no association was found between small community residence and increased HPV infection or penile cancer risk, contradicting studies from South America, Central America, and Asia [3,48,50,51]. This may be due to enhanced sexual education and protective measures against sexually transmitted infections in Poland. Our investigation showed that 49 patients (32.07%) had both phimosis and small community residence. Additionally, 13 (8.5%) patients were beekeepers living in rural areas, though no significant correlation existed between this occupation and increased penile cancer risk. While this is the first study examining beekeeping as a potentially associated with penile cancer, it may be coincidentally related to rural lifestyle, warranting further research.

### 5.4. The Role of Tobacco in Penile Cancer

Smoking is a well-established risk factor for numerous malignancies and penile cancer development. Initial reports on smoking’s effects on penile cancer were published in 1987 by Dan Hellberg and colleagues, who demonstrated, in 244 Swedish men, a correlation between smoking and penile cancer, independent of phimosis and balanitis. When smokers were categorized by cigarettes consumed per day, Hellberg et al. observed a dose–response relationship, with individuals smoking more than 10 cigarettes/day showing significantly higher risk than light smokers (1–10 cigarettes a day) (χ^2^ = 5.43, *p* = 0.02). The relative risk of developing penile cancer for heavy smokers was 1.88 (95% CI = 1–10 to 3–19) compared to light smokers and 2.22 compared to non-smokers [12]. Similarly, Harish et al. identified cigarette smoking as a significant risk factor for penile cancer development in individuals who consumed more than 10 cigarettes daily (OR = 2.143, *p* < 0.001) and those who smoked for over 5 years (OR = 1.433, *p* = 0.006). Cigarette smoking, tobacco chewing, and their combination were associated with elevated penile cancer risk (OR = 3.396, *p* < 0.001) [56]. The incidence of in situ and invasive penile cancer among men who had smoked was 2.4 times higher than in non-smokers [23]. A case-control study of 137 men with in situ (*n* = 75) and invasive (*n* = 62) penile cancer showed that smoking was associated with a 4.5-fold increased risk (95% CI = 2.0–10.1) of invasive penile cancer, although no association was found with in situ cancer [22]. In a study by Chalya et al., 77.1% of penile cancer cases were linked to smoking [57], with current smokers showing more pronounced effects than former smokers [58]. In our investigation, we observed that 59 of 153 patients with penile cancer (38.56%) were current smokers. The data demonstrated a significant association between smoking and mortality risk, with smokers exhibiting a two-fold higher risk and significantly reduced OS than non-smokers (*p* = 0.047). The analysis of OS rates, which includes other morbidities, revealed significant differences between the OS for smokers and non-smokers (Figure 5).

The association between tobacco use and other squamous cell carcinomas, e.g., those of the lungs and cervix, is well known. Cigarette smoking has been shown to impair immune function in the female cervical epithelium by reducing the local number of Langerhans cells. This alteration may influence the susceptibility to HPV infection [58].

The precise mechanism by which smoking promotes penile cancer development is not understood. However, this influence is dose-dependent and associated with genital squamous cell carcinomas. The accumulation of nicotine and cotinine in genital secretions may be important; tobacco products concentrate in smegma, potentially making it carcinogenic, particularly with phimosis [57,59].

### 5.5. The Role of HPV in Penile Carcinogenesis: From Infection to Prognosis

Human papillomavirus (HPV) is considered the etiological factor in approximately 5% of all cancer cases globally, affecting both sexes [60]. The magnitude of this issue is emphasized by the fact that HPV is the most prevalent sexually transmitted virus worldwide [61]. Bruni et al. indicated that nearly one-third of men globally are infected with at least one genital HPV type and approximately one-fifth with one or more high-risk HPV types [62]. In the context of penile cancer, it is estimated that 30–50% of cases are associated with HPV infection, particularly the high-risk types, 16 and 18 [2]. Furthermore, a strong correlation is evidenced by the detection of HPV genetic material in 70–100% of cases of penile intraepithelial neoplasia (PeIN), which represents a precancerous lesion [2,63].

In our study of 153 cases, only 22 (14.38%) were found to be associated with HPV infection. These findings are consistent with those reported by Borque-Fernando et al., who analyzed penile cancer in the Spanish National Registry. In their study, 155 of 228 patients were tested for HPV infection, with only 28 patients (18.1%) being HPV-positive [26]. Contrasting results were presented by Gu et al., who examined 340 penile cancer specimens obtained from five tertiary hospitals in China, diagnosed from 2006 to 2017. The investigators utilized the PCR-RBD HPV test method to detect HPV genetic material in 166 cases (48.8%) and subsequently assessed p16INK4a expression to confirm HPV infection, revealing its overexpression in 45.6% of cases [64]. Similarly, Coelho et al. observed 66.9% (91/136) HPV-positive penile cancer specimens in a study from northeast Brazil [3]. In a study conducted in the same region of Brazil, Vieira et al. identified 62% (72/116) HPV-positive penile cancer cases [50]. The significant disparities in the results may be attributed to the varying epidemiology of HPV across different countries and continents.

The impact of HPV infection on penile cancer prognosis remains contentious. Mustasam et al. reviewed the literature on p16^INK4a^ protein expression and HPV DNA detection’s impact on disease-specific survival (DSS), overall survival (OS), and disease-free survival (DFS). Their study found beneficial effects of p16INK4a protein on DSS, OS, and DFS, with HPV DNA positively correlating with DFS and DSS [65]. Lont et al. showed better 5-year DSS in HPV-positive patients versus HPV-negative patients (93% vs. 78%; *p* = 0.03) [66]. However, Bezerra et al. found no significant differences between HPV-negative and HPV-positive patients in lymph node metastasis (*p* = 0.386) and 10-year survival rates (68.4% vs. 69.1%; *p* = 0.83) [67]. In our study, we observed an almost six-fold (HR = 0.15) reduction in the risk of death in HPV-positive patients; however, this result did not reach statistical significance (*p* = 0.063) (Figure 6).

### 5.6. Centralization of Penile Cancer Care

The centralization of rare diseases like penile cancer offers advantages, including the development of experienced multidisciplinary teams, improved survival, higher rates of minimally invasive nodal staging, increased organ-sparing surgery, and comprehensive pathology reports. This approach creates opportunities for research and trials. However, disadvantages include the lack of local patient support and low experience among urologists outside reference centers. The main patient concern is distance from the reference center [68,69,70].

Only one study has examined the distance from referral penile cancer centers. Pecoraro et al. found that after centralization in 2015, the median travel distance increased to 55.4 (23.8–73.6) km compared to 37.5 (10.1–69.4) km in 1989–2014. The relationship between distance and prognosis remains unstudied [69]. In our study, the median distance was 62 km (mean: 104.54; SD ± 120 km), with 26.14% (*n* = 40) of patients considered as being far from the center. A slight negative correlation existed between advanced age (>71 years) and distance (>147 km) (r = −0.27; *p* < 0.05); however, no association was found between cancer advancement and prognosis.

### 5.7. Limitations and Future Research Directions

The strength of this study lies in its comprehensive analysis of various factors associated with penile cancer and their impacts on prognosis in a Central European country. Our study contributes to the knowledge on penile cancer in Central European populations, specifically Poland, addressing a significant knowledge gap, given the rarity of penile cancer cases reported in this geographic region and the sociodemographic context. The limitations of our study include its retrospective design and the reliance on self-reported assessments of clinical and behavior features. This issue is particularly relevant in the context of smoking, where data on pack-years are unavailable, and phimosis, where information regarding its duration or whether it is primary or secondary is lacking. Furthermore, the prevalence of HPV infection in penile cancer tumors was evaluated using p16 immunohistochemistry. This method serves as a valuable surrogate marker for transcriptionally active high-risk HPV; however, it is neither fully specific nor fully sensitive. Additionally, the follow-up period, with a median duration of 27 months (mean: 37.61 months; range: 4–159 months), may influence the outcomes, particularly concerning overall survival (OS) or cancer-specific survival (CSS). Furthermore, the absence of a control group in our study design prevented us from establishing causal relationships between the identified clinical and lifestyle characteristics and penile cancer, thus limiting our findings to associative observations.

Future multicenter prospective studies with larger sample sizes, control groups, and extensive follow-up (>5 years) should be conducted to establish strong causal relationships between the identified risk factors and their impact on prognosis. Additionally, it is essential to gather information regarding pack-years, the duration of the symptoms of phimosis, grade, and whether the phimosis is primary or secondary. Furthermore, in the context of HPV infection, in addition to immunohistochemical testing for p16, molecular tests, such as polymerase chain reaction (PCR) or in situ hybridization, should be conducted. Moreover, future research should focus on exploring the underlying mechanisms of these risk factors and developing comprehensive screening programs to improve the early detection and patient outcomes of penile cancer.

## 6. Conclusions

Phimosis was the most frequently observed clinical characteristic in our cohort and was correlated with a more advanced penile cancer tumor stage. Smoking was associated with worse patient survival, while HPV-positive patients demonstrated a trend toward lower mortality compared to HPV-negative patients.

## Figures and Tables

**Figure 1 cancers-17-02140-f001:**
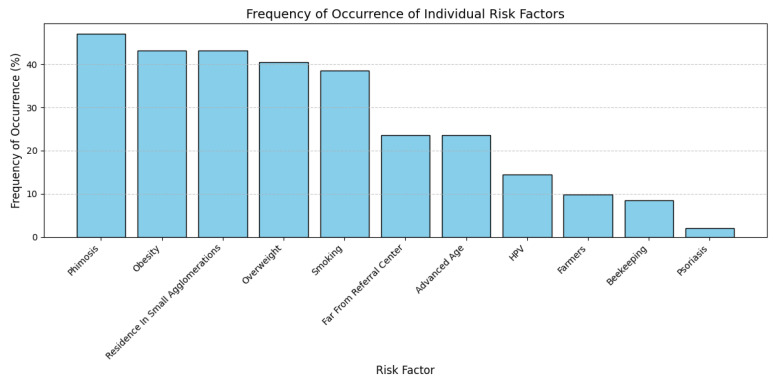
Distribution of clinical and lifestyle characteristics among penile cancer patients.

**Figure 2 cancers-17-02140-f002:**
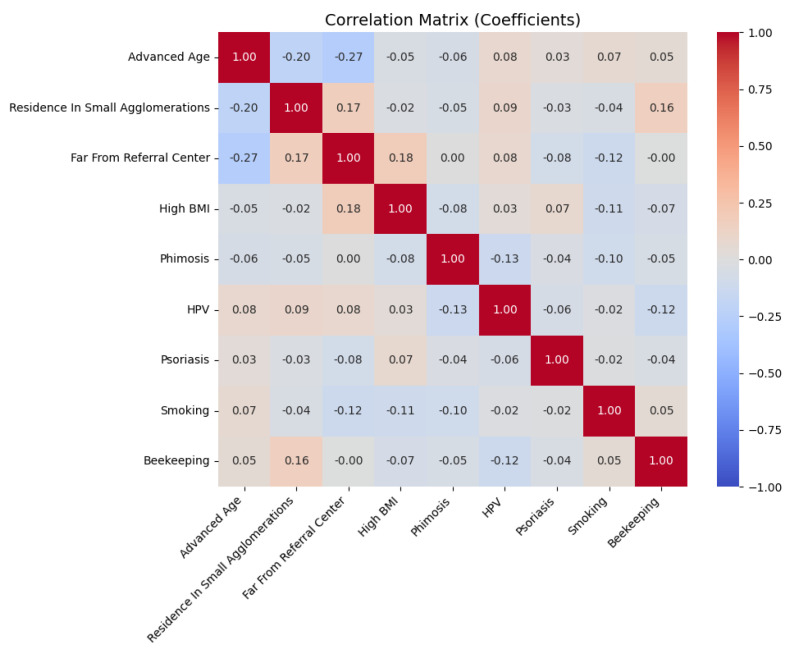
Matrix of interrelationships among various factors.

**Figure 3 cancers-17-02140-f003:**
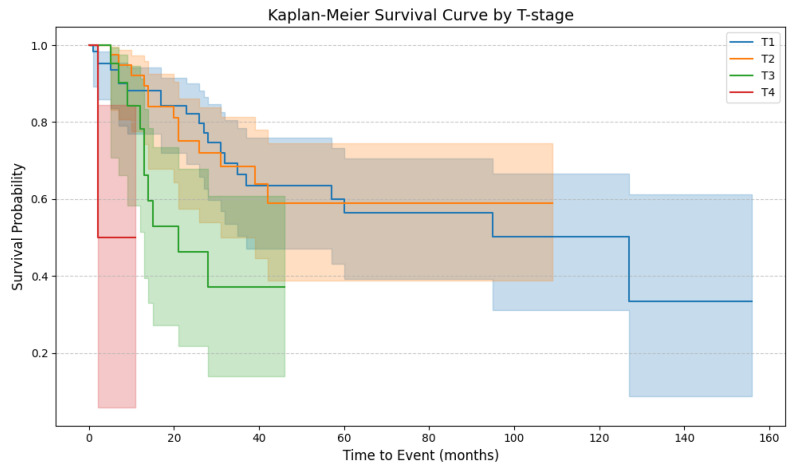
The Kaplan–Meier plot with survival curves for different local tumor stages (T1–T4). The shaded regions surrounding the curves represent the 95% confidence intervals. T1, blue line; T2, orange line; T3, green line; T4, red line (T1 vs. T2, *p* = 0.93; T2 vs. T3, *p* = 0.03; T3 vs. T4, *p* = 0.01) (T1–T4, *p* < 0.005).

**Figure 4 cancers-17-02140-f004:**
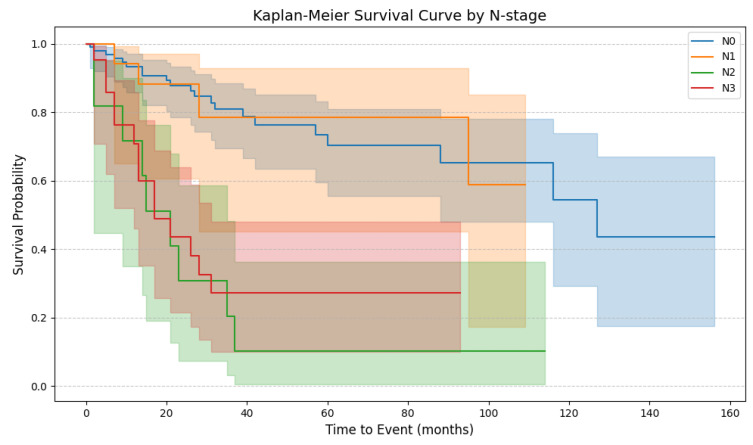
The Kaplan–Meier plot with survival curves (OS) for lymph node stages (N0–N3). The shaded regions surrounding the curves represent the 95% confidence intervals. N0, blue line; N1, orange line; N2, green line; N3, red line (N0 vs. N1, *p* = 0.97; N1 vs. N2, *p* < 0.005; N2 vs. N3, *p* = 0.54) (N0–N3, *p* < 0.005).

**Figure 5 cancers-17-02140-f005:**
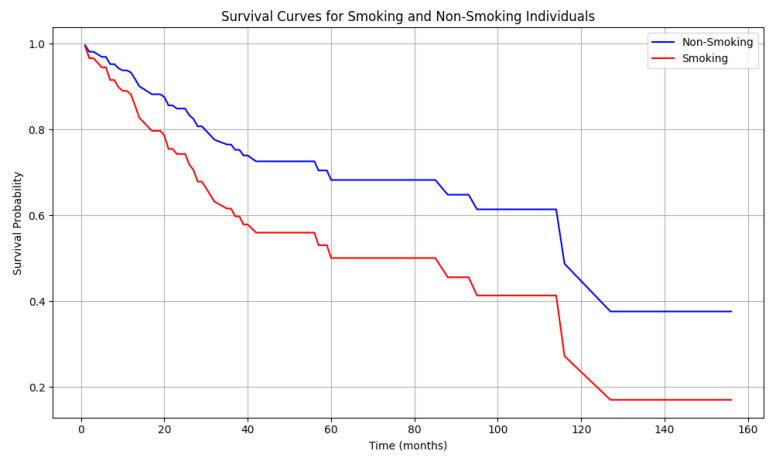
The Cox plot shows the survival functions (OS) for two groups of patients: non-smokers (blue line) and smokers (red line). The horizontal axis (X) represents survival time in months, and the vertical axis (Y) represents the probability of survival (*p* = 0.047).

**Figure 6 cancers-17-02140-f006:**
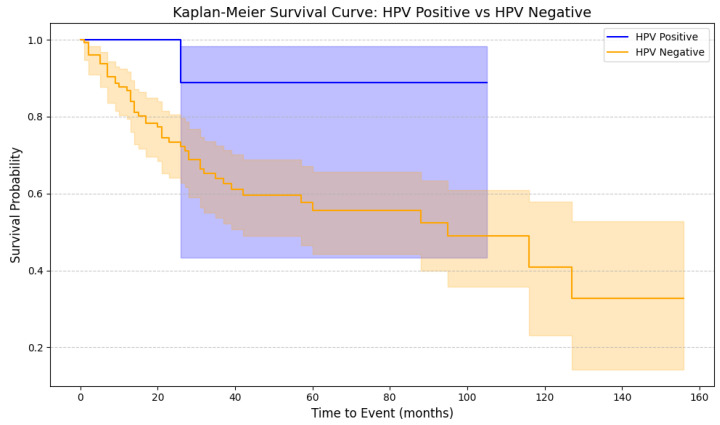
Kaplan–Meier survival curves comparing overall survival between HPV-positive and HPV-negative patients. The survival probabilities were plotted over time (in months), with shaded areas representing 95% confidence intervals for each group. The blue curve indicates the HPV-positive group, while the orange curve represents HPV-negative patients. The plot reveals that patients with HPV positivity tend to have higher survival probabilities across the observed period. This survival analysis was conducted using the Python lifelines package, applying Kaplan–Meier estimators for group-wise survival distribution comparison. A statistically significant difference between the survival curves was observed (*p* = 0.03).

**Table 1 cancers-17-02140-t001:** Demographic and histopathological characteristics of patients.

Patients	Criteria		Count	Percent [%]
**Age (Years)**	Minimum		30	-
	Maximum		87	
	Median		64	
**BMI [kg/m^2^]**	Minimum		16	-
	Maximum		38	
	Median		29	
**Religious faith**	Jehovah’s Witnesses		3	1.96
	Catholicism		150	98.04
**Circumcised (before diagnosis)**	Yes		0	0
	No		153	100
**Residential distribution**	Rural areas		42	27.45
	Small urban centers		24	15.95
	Medium-sized cities		41	26.80
	Large metropolitan areas		46	30.07
**TNM—local stage**	Tis		22	14.38
	pT1a		45	29.41
	pT1b		20	13.07
	pT2		38	24.84
	pT3		23	15.03
	pT4		5	3.27
**TNM—lymph nodes**	pN0		99	64.71
	pN1		20	13.07
	pN2		11	7.19
	pN3		23	15.03
**TNM—distant metastases**	M0		150	98.04
	M1		3	1.96
**Grading**	PeIN		22	14.38
	G1		35	22.88
	G2		68	44.44
	G3		28	18.30
**Histopathological type**	Squamous cell carcinoma		147	96.08
	Verrucous squamous cell carcinoma		6	3.92
**Surgical treatment**	Circumcision		20	13.07
	Wide local excision		12	7.84
	Glansectomy with reconstruction using a split-thickness skin graft		55	35.95
	Partial penectomy with reconstruction using a split-thickness skin graft		49	32.03
	Partial penectomy without reconstruction		10	6.54
	Total penectomy		7	4.57
**Radiotherapy**	Primary tumor radiotherapy		2	1.3
	Adjuvant radiotherapy		6	3.91
**Chemotherapy**	Neoadjuvant chemotherapy (TIP *)		7	4.6
	Adjuvant chemotherapy	TIP *	32	20.92
		PF **	10	6.54
	Immunotherapy		8	5.23

* TIP—paclitaxel, cisplatin and ifosfamide; ** PF—cisplatin and paclitaxel.

**Table 2 cancers-17-02140-t002:** Univariate and multivariate analyses using a Cox proportional hazard regression model for overall survival.

Parameter	Univariate Analysis			Multivariate Analysis			
	HR	95% CI	*p*-Value	HR	95% CI	*p*-Value	Statistical Power
Age (centered)	1.02	1.02–1.05	0.08787	1.02	1.00–1.05	0.098	0.38
Smoking	1.76	1.00–3.11	0.052	1.81	1.01–3.25	**0.047 ***	0.51
HPV	0.15	0.02–1.11	0.063	0.15	0.02–1.11	0.063	0.46
BMI	0.19	0.03–1.08	0.060	-	-	-	-
Phimosis	0.79	0.44–1.42	0.428	-	-	-	-
Beekeeping	0.39	0.09–1.62	0.195	-	-	-	-
Residence in small agglomerations	0.97	0.77–1.23	0.808	-	-	-	-
Psoriasis	2.40	0.74–7.77	0.143	-	-	-	-
Lymph node stages	3.36	1.86–6.06	<0.001	2.10	1.09–4.06	**0.027 ***	0.60
Tumor grade	2.10	1.46–3.03	<0.001	1.85	1.23–2.78	**0.003 ***	0.85

* Significant results (*p* < 0.05) are given in bold. Abbreviations: HR—hazard risk ratio; CI—confidence interval.

## Data Availability

The original contributions presented in this study are included in the article and Appendix A. Further inquiries can be directed to the corresponding author.

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
