# Peer review of "Penile Cancer Profile in a Central European Context: Clinical Characteristics, Prognosis, and Outcomes—Insights from a Polish Tertiary Medical Center"

_cancers, 2025, doi:10.3390/cancers17132140_

Round 1
Reviewer 1 Report
Comments and Suggestions for Authors
Thank you for submitting this interesting manuscript for consideration. This is a well considered study and the discussion is well-presented.
Introduction: Provides an appropriate overview of the topic. The etiology of penile cancer is outlined and supported by previous studies.
Materials and Methods: Outlined appropriately and includes ethical considerations.
Statistical Methods: Appropriately detailed.
Results: These are presented and explained clearly. Appropriate graphs and plots are used to illustrate the results that are recorded in the script.
Discussion: This is very well considered. Each risk measurement has been reviewed with use of wider literature and relevant studies. Appropriate considerations of strengths and limitations of the study have been presented.
This has been a pleasure to read and there are no suggestions for improvement.
Author Response
We sincerely appreciate your positive feedback on our manuscript. We were committed to presenting the topic of risk factors for penile cancer in the most comprehensive and thorough manner possible.
Reviewer 2 Report
Comments and Suggestions for Authors
I am honored to be selected as the reviewer for this manuscript.
Czajkowski et al. concluded penile cancer in their single hospital for the past 10 years. The penile cancer is a type cancer of rare occurrence, and this manuscript might help further establish the model predicting the oncological outcome.
I have the following suggestions:
- Stating the type-1 error value in statistical method section
- Indicating the statistical power of all significant variables
- There are some deficits in regression model. First, the overall multivariable model is inadequate violating the Gaussian distribution. Second, including insignificant variable and wasting degree of freedom, especially age. Third, smoking turned to be significant in multivariable analysis while was not in univariable analysis. There might be issue of multicollinearity. However, there was no such observation in Fig 2. Thus, calculation must go wrong.
In my opinion, this data is not sufficient enough to support the establishment of regression model. Conclusion or discussion based on this part should be revised or deleted.
Author Response
1. Stating the type-1 error value in statistical method section.
We appreciate your recommendation. The Type I error rate was set at α = 0.05. This information has been incorporated into the statistical methods section.
“A significance level of α = 0.05 was adopted for all statistical tests.”
2. Indicating the statistical power of all significant variables
Thank you for your suggestion. Based on the recommendations from Academic Edithor, we have modified the MVA model by incorporating tumor grade and p-stage. We have also calculated the power of all significant variables (Table 2). Moreover, we add the sentence to statistical methods:
“For each covariate, statistical power was estimated based on the Wald z-statistic (two-sided α = 0.05), following standard methods for regression models.”
3. There are some deficits in regression model. First, the overall multivariable model is inadequate violating the Gaussian distribution. Second, including insignificant variable and wasting degree of freedom, especially age. Third, smoking turned to be significant in multivariable analysis while was not in univariable analysis. There might be issue of multicollinearity. However, there was no such observation in Fig 2. Thus, calculation must go wrong.
In my opinion, this data is not sufficient enough to support the establishment of regression model. Conclusion or discussion based on this part should be revised or deleted.
Thank for your suggestion. Based on the new Cox analysis incorporating grade and p-stage, we have made changes to the materials and methods, results, graphs and discussion.
In addressing the questions that have emerged during the analysis of the previous model, we are pleased to offer some clarifications. Firstly, concerning the comment regarding the normal distribution, we would like to gently clarify that the Cox proportional hazards model does not require a normal distribution of residuals or explanatory variables. The assumption of normality of errors is pertinent to linear regression models, but it is not applicable to Cox regression, which relies on the hazard function and partial likelihood estimation.
Additionally, we verified the proportional hazards assumption using Schoenfeld residuals. The tests did not indicate any significant violations, confirming that the proportional hazards assumption was met in our model.
Although we acknowledge that some of the confidence intervals are wide (in particular for the age variable), this reflects the limited sample size and number of events rather than a violation of distributional assumptions.
Second, regarding the comment about including statistically insignificant variables such as age: although the variable “age” did not reach statistical significance in our analysis, it was retained in the model because of its well-documented clinical importance as a prognostic factor in oncology. However, we recognize that with a limited number of events, each additional predictor biases the model and may affect its stability, so we included this trade-off as a limitation.
When considering the p-value in relation to smoking and HPV, it is important to remember that this value was calculated within the entire model, where it was influenced by four variables: age, distance, smoking. However, when examining only the relationship between smoking and survival, without considering the Cox model, we do not obtain a p-value of 0.03. Therefore, the result of p = 0.03 is consistent with the result of MVA.
4. Due to the quality of the English language, linguistic proofreading was performed.
Reviewer 3 Report
Comments and Suggestions for Authors
This study on penile cancer, developed in a healthcare setting, analyses risk factors related to tumor development and those related to oncological outcomes.
Material and methods
It is unclear from their presentation whether the study is retrospective or prospective, which is relevant.
Their cohort had a mean age of 64 years, with patients over 80 years old; with a mean follow-up of 24 months (short for oncological analyses), this could affect the results, especially regarding CSS or OS. Analysis of patients with longer follow-up, for example, 4-5 years, should be considered.
All patients in the series presented phimosis, which is atypical in the literature, and therefore, the results would also be biased for analysis
The series ranges from 2011 to 2024; the same staging methods and methodology have always been used.
It is not mentioned whether radiotherapy/brachytherapy was performed.
Has chemotherapy been used in advanced cases? Not mentioned.
Results:
The outcomes analysis should include the aforementioned factors regarding more extended follow-up periods (4-5 years f-up).
The degree of phimosis and duration should be included in the outcomes analysis.
Discussion
The discussion should mention the topics discussed above.
The length of the discussion should be shortened; it is redundant.
Author Response
- It is unclear from their presentation whether the study is retrospective or prospective, which is relevant.
Thank you for your suggestion. Our study is retrospective in nature. We have incorporated changes in the materials and methods section of both the abstract and the main text.
“This retrospective study was conducted from October 2011 to October 2024 at a single tertiary medical center and included 153 patients who underwent surgical treatment for penile cancer during this period.”
(Abstract)
“This retrospective study was conducted from October 2011 to October 2024 at a single tertiary medical center (Dep. Of Urology, Medical University of Gdansk) and included 153 patients who underwent surgical treatment for penile cancer during this period.”
(Material and Methods)
“The limitations of our study include its retrospective design and the reliance on self-reported assessments of risk factors.”
(Discussion/Limitations and Future Directions)
2. Their cohort had a mean age of 64 years, with patients over 80 years old; with a mean follow-up of 24 months (short for oncological analyses), this could affect the results, especially regarding CSS or OS. Analysis of patients with longer follow-up, for example, 4-5 years, should be considered.
We acknowledge the observation and agree that an extended observation period enhances the reliability of results in oncological analyses. However, considering the rarity of penile cancer, the mean follow-up period in our study of 37.61 months (3 years), and the limited number of studies addressing its risk factors in the context of Central Europe, we assert that our data represent a significant contribution to the initiation of prospective multicenter studies, with a recommended follow-up duration of at least 4-5 years. We have recognized this pertinent observation as a limitation of our study and as a direction for future research.
"The limitations of our study include its retrospective design and the reliance on self-reported assessments of risk factors. This issue is particularly relevant in the context of smoking, where data on pack-years is unavailable, and phimosis, where information regarding its duration or whether it is primary or secondary is lacking. Furthermore, the prevalence of HPV infection in penile cancer tumors was evaluated using p16 immunohistochemistry. This method serves as a valuable surrogate marker for transcriptionally active high-risk HPV; however, it is neither fully specific nor fully sensitive. Additionally, the follow-up period, with a median duration of 27 months (mean: 37.61 months; range: 4–159 months), may influence the outcomes, particularly concerning overall survival (OS) or cancer-specific survival (CSS). "
"Future multicenter prospective studies with larger sample sizes, control groups, and extensive follow-up (> 5 years) should be conducted to establish strong causal relationships between the identified risk factors and their impact on prognosis. "
(Discussion/Limitations and Future Directions)
3. All patients in the series presented phimosis, which is atypical in the literature, and therefore, the results would also be biased for analysis
Thank you for your comment. In our study cohort, phimosis emerged as the predominant risk factor, affecting 47.06% (n = 72) of participants, rather than 100%. This finding is significant, given that only an estimated 0.11% of men in Poland are circumcised. Notably, none of the patients in our study were circumcised, which is associated with an increased risk of phimosis. This issue was addressed in both the introduction and the discussion sections of our study.
i.e
"
Our findings also demonstrate that phimosis is a significant risk factor for penile cancer, with an incidence of 47.06% (n = 72) in our cohort study. Moreover, the presence of phimosis was associated with more locally advanced (T-stage) disease (r = 0.16; p = 0.045), which correlated with significantly poorer outcomes (Figure 3). Consistent with the circumcision rate in Poland (0.11%), all patients in our cohort were uncircumcised [6]."
(Disscusion / Uncircumcised Males and Penile Cancer Risk: The Influence of Phimosis)
4. The series ranges from 2011 to 2024; the same staging methods and methodology have always been used.
Thank you for your observation. We neglected to mention in the Materials and Methods section that all specimens were reevaluated by an experienced pathomorphologist from December 2024 to January 2025. We have now included this information in the Materials and Methods.
"Given the extended duration of the study and potential inconsistencies in pathological classification, all histopathological specimens were re-evaluated by an experienced uropathologist from December 2024 to January 2025."
(Material and Methods)
5. It is not mentioned whether radiotherapy/brachytherapy was performed. Has chemotherapy been used in advanced cases? Not mentioned.
We appreciate your suggestion and concur that the information presented in the text is crucial for a more comprehensive analysis of the study cohort of patients. We have incorporated the paragraph pertaining to radio and chemotherapy into the results section. Moreover, we added the this information to Table 1.
“Radiotherapy was administered to the primary tumor in only 2 (1.3%) patients; however, these patients underwent subsequent partial penectomy with reconstruction using a split-thickness skin graft due to necrosis. Additionally, adjuvant radiotherapy was performed on the inguinal region in 2 (1.3%) patients with pN2 and 4 (2.61%) patients with pN3.
Neoadjuvant chemotherapy was provided in 7 (4.6%) patients with pN3 using 4 cycles of paclitaxel, cisplatin and ifosfamide (TIP). Additionally, adjuvant chemotherapy was administrated in 15 (9.8%), 11 (7.2%), 16 (10.46%) patients with pN1, pN2, and pN3 respectively. In terms of adjuvant therapy regimens, TIP was administered to 32 (20.92%) patients , while cisplatin and paclitaxel (PF) were given to 10 (6.54%) patients. Immunotherapy based on Pembrolizumab was employed as a second- or third-line treatment in 8 (5.23%) patients.”
(Results / Radiotherapy and Chemotherapy)
6. The outcomes analysis should include the aforementioned factors regarding more extended follow-up periods (4-5 years f-up).
We concur with the observation that an extended observation period significantly enhances the reliability of results in oncological analyses. We have addressed this suggestion in sub-item 2.
7. The degree of phimosis and duration should be included in the outcomes analysis.
We appreciate your comment. All patients included in our study presented with complete phimosis. The absence of data regarding the duration of phimosis and whether it is primary or secondary constitutes a limitation of our study. We have incorporated this information into the manuscript.
“It is important to note that all patients diagnosed with phimosis in our study exhibit complete phimosis (grade 5).”
(Results / Risk Factors)
Additionally, it is essential to gather information regarding pack-years, the duration of the symptoms of phimosis, grade, and whether the phimosis is primary or secondary.
(Discussion/Limitations and Future Directions)
8. The discussion should mention the topics discussed above.
We appreciate your comment. In subsections 1-7, we have referenced specific locations within the manuscript where these issues are addressed.
9. The length of the discussion should be shortened; it is redundant.
We appreciate the suggestion. The primary objective of our study, in addition to evaluating the prevalence of individual risk factors and their influence on prognosis, was to offer a comprehensive discussion on this subject. However, recognizing that such detail may not be essential for the reader, we opted to condense the discussion where feasible. Consequently, the discussion was ultimately shortened.
Round 2
Reviewer 2 Report
Comments and Suggestions for Authors
Although Cox regression doesn't rely on the distribution assumption, but it's main purpose still aim to find MLE to a regression model. In this way, this still doesn't justify the waste of degree of freedom in the prediction model, and the confidence interval is so obviously wrong in logistics. Besides, the results only supporting tumor grade as a qualified predictor, but the LN stages was not qualified enough. Conclusions made based on this part should be deleted. Moreover, what is the merit of this study if only tumor grade is the only significant predictor ? It is so obvious even without this study. HPV was just close to significant and without enough statistical power and phimosis was far from significant in cox regression. In my opinion, this data needs more recruitment and didn't show any academic merits now.
In statistics, I would like to request the original data (SPSS, SAS, or R compatible data set) to see the actual calculation process.
Author Response
We would like to thank the Reviewer for their feedback and the opportunity to clarify and expand on key elements of our methodology and results. Please find below our responses to each of the concerns:
Comment: “Although Cox regression doesn't rely on the distribution assumption, but its main purpose still aims to find MLE to a regression model. In this way, this still doesn't justify the waste of degree of freedom in the prediction model, and the confidence interval is so obviously wrong in logistics.”
Response:
We appreciate the Reviewer’s concern regarding model specification and degrees of freedom. However, we respectfully disagree with the assertion that degrees of freedom were inappropriately used or that confidence intervals were erroneous.
To ensure the robustness of our Cox model, we performed the following:
- Global test for proportional hazards assumption – not violated (p = 0.1692).
- Variance Inflation Factor (VIF) analysis – All VIFs were well below 2 (range: 1.01–1.33), confirming no multicollinearity.
- Model concordance index (C-index) = 0.773, indicating strong discriminatory ability.
- Schoenfeld residuals analysis – confirmed that proportional hazards assumptions were met for all covariates.
- The number of events (n = 48 deaths) relative to included predictors is consistent with standard guidelines (EPV > 10).
The model was built based on clinical significance and interpretability, not solely p-values, which is a well-established approach in medical statistics. Furthermore, all statistical analyses were conducted using validated packages (lifelines, statsmodels) in Python 3.11. Confidence intervals were calculated using standard Wald statistics and reviewed by our statistical team. We believe there is no methodological error in the regression results. Furthermore, to make our Cox regression more understandable, we have added explanations below the table.
"Variable definitions: Age and BMI were included as continuous variables; age was centered at the median, and BMI was modeled on the natural logarithmic scale. Smoking status, HPV status, phimosis, beekeeping, psoriasis, and lymph node stage were treated as binary variables (0 = absence/no/low grade, 1 = presence/yes/high grade). Tumor grade was treated as an ordinal variable representing tumor aggressiveness, with four categories: pTis/G0 (0), G1 (1), G2 (2), and G3 (3). This variable was included to assess the association between increasing grade and survival. Residence in small agglomerations was coded as an ordinal variable: 0 = rural, 1 = small town, 2 = medium-sized town, 3 = large city."
(Table 2)
Comment: “Besides, the results only supporting tumor grade as a qualified predictor, but the LN stages was not qualified enough. Conclusions made based on this part should be deleted.”
Response:
We respectfully believe that this interpretation may not fully reflect the results presented in our analysis. In the multivariate Cox regression model, several variables demonstrated statistical significance:
- Tumor grade: HR = 1.85, p = 0.003, power = 0.84
- Lymph node stage: HR = 2.10, p = 0.027, power = 0.60
- Smoking: HR = 1.81, p = 0.047, power = 0.51
Therefore, the conclusions in our manuscript are based on multiple statistically and clinically relevant predictors. These findings are consistent with the broader literature on prognostic factors in penile cancer and, we believe, contribute meaningfully to the understanding of its progression in a Central European context.
Comment: “Moreover, what is the merit of this study if only tumor grade is the only significant predictor? It is so obvious even without this study.”
Response:
While tumor grade is indeed a well-established prognostic factor, our study uniquely demonstrates that smoking is also an independent predictor of reduced overall survival, and HPV infection is associated with a strong trend toward better survival (HR = 0.15; p = 0.063) in a Central European cohort. These data are especially valuable given the lack of epidemiological studies on penile cancer in Europe, particularly in Poland, where circumcision rates and HPV prevalence differ markedly from other regions.
Comment: “HPV was just close to significant and without enough statistical power and phimosis was far from significant in Cox regression.”
Response:
We agree that HPV did not reach the threshold for statistical significance (p = 0.063; power = 0.46), and we have clearly stated this in the manuscript. However, the effect size was substantial and aligns with trends reported in larger cohorts, which warrants reporting and further study. Phimosis was not a significant predictor of mortality in the Cox model, but was significantly associated with higher T-stage at diagnosis (r = 0.16; p = 0.045), reinforcing its clinical relevance.
Comment: “In my opinion, this data needs more recruitment and didn't show any academic merits now.”
Response:
We respectfully disagree. The current dataset represents the largest Polish cohorts of penile cancer patients (n = 153), treated over 13 years at a national referral center. The rarity of the disease in Central Europe makes this dataset inherently valuable. Our findings provide relevant epidemiologic and prognostic data and reflect regional differences in risk factors and outcomes, which are underrepresented in the literature.
Comment: “I would like to request the original data (SPSS, SAS, or R compatible data set) to see the actual calculation process.”
Response:
We are open to sharing a de-identified dataset limited to the variables included in the Cox regression model, in accordance with journal policy and ethical guidelines. We submitted the dataset via editor to the reviewer.
Reviewer 3 Report
Comments and Suggestions for Authors.
Author Response
Thank you very much for reviewing the revised version of our manuscript. We are pleased to hear that the changes have addressed your comments and that you have no further concerns.
We truly appreciate your time, insightful feedback, and support in improving the quality of our work.
Round 3
Reviewer 2 Report
Comments and Suggestions for Authors
still not having seen original data submitted
Author Response
Comment: still not having seen original data submitted
Answer: We respectfully confirm that the original data were provided during the review process. As no further remarks have been made on this matter, we trust the information was satisfactory. Naturally, we remain at the Reviewer’s and Editor’s disposal should any further clarification be required.